# SpeAr: A Spectral Approach for Zero-Shot Node Classification

**Ting Guo[1], Da Wang[2], Jiye Liang[2]\*, Kaihan Zhang[1], Jianchao Zeng[1]**
1. Data Science and Technology, North University of China, Taiyuan, China.
2. School of Computer and Information Technology, Shanxi University, Taiyuan, China.

## Abstract

Zero-shot node classification is a vital task in the field of graph data processing, aiming to identify nodes of classes unseen during the training process. Prediction bias is one of the primary challenges in zero-shot node classification, referring to the model's propensity to misclassify nodes of unseen classes as seen classes. However, most methods introduce external knowledge to mitigate the bias, inadequately leveraging the inherent cluster information within the unlabeled nodes. To address this issue, we employ spectral analysis coupled with learnable class prototypes to discover the implicit cluster structures within the graph, providing a more comprehensive understanding of classes. In this paper, we propose a **Spe**ctral **Ap**p**r**oach for zero-shot node classification (SpeAr). Specifically, we establish an approximate relationship between minimizing the spectral contrastive loss and performing spectral decomposition on the graph, thereby enabling effective node characterization through loss minimization. Subsequently, the class prototypes are iteratively refined based on the learned node representations, initialized with the semantic vectors. Finally, extensive experiments verify the effectiveness of the SpeAr, which can further alleviate the bias problem.

## 1 Introduction

Graph data is widely used to reveal interactions between various entities, such as citation networks [1], social networks [2], recommendation systems [3], etc. In graph data structures, various entities are abstractly represented through the form of nodes, while their complex relationships are precisely depicted through the connections of edges. Node classification [4, 5], an essential task in graph data analysis, focuses on predicting labels for unlabeled nodes by harnessing the relational structure of the graph and a select subset of labeled nodes. The advent of graph neural network (GNN) technology, as delineated in [6, 7, 8], represents a monumental stride in the domain of node classification. This development has accelerated advancements in the analysis of graph-structured data.

Nevertheless, the ongoing dynamics of real-world networks, characterized by the continual emergence of new nodes and edges, facilitate the discovery of new classes, thereby introducing a series of innovative challenges for node classification tasks [9]. Traditional GNN-based node classification models struggle to handle the identification of new classes in dynamic and open environments. For instance, as citation networks continue to grow, the emergence of new research papers promotes the rise of new academic fields. A key challenge is the effective integration of newly emerging nodes into the pre-existing foundational classes or nascent ones currently in development. Therefore,

---

\*Corresponding author. Email: ljy@sxu.edu.cn. This work is supported by the National Science and Technology Major Project (2020AAA0106102), the National Natural Science Foundation of China (62376141), and the Natural Science Foundation of Shanxi Province, China (202203021222075).

constructing a model that can recognize nodes of unseen classes offers significant practical utility for the analysis of graph data.

Zero-shot node classification (ZNC) [10] presents a potent strategy for tackling the recognition of unseen classes that are absent from the labeled nodes. It endeavors to harness external categorical knowledge (e.g. semantic vectors) and comprehensive graph data (e.g. node attributes and structural information), to classify nodes belonging to unseen classes. In ZNC, prediction bias is a significant issue, where models incorrectly predict unseen class nodes as seen classes. Several studies concentrate on constructing optimized node representations and ensuring their alignment with semantic vectors in the latent space. This alignment is achieved by minimizing the distance between nodes and their corresponding class semantic vectors. DGPN [10] captures the local and global embedding of graph nodes, devising a distance loss function that bridges the gap between graph representations at varying scales and semantic vectors. MVE [11] underscores the insufficiency of modeling node features from a single view in describing the comprehensive essence of a node, thus advocating for a multi-view approach to augment node features. In addition, the utilization of inter-class relationships is also crucial in ZNC. DBiGCN [12] incorporates class relationships into the node embedding process to facilitate the model's transfer to unseen classes. GraphCEN [13] enhances model adaptation to unseen classes through a dual contrastive loss mechanism that captures node-class dependencies.

Existing research predominantly leverages external knowledge, such as semantic vectors and the category relationships stemming from them, to mitigate prediction bias. However, most methods insufficiently exploit the cluster information inherent within unlabeled nodes. Strategically leveraging this latent information can reveal hidden class cluster structures, thereby facilitating a comprehensive understanding of the categories present within the graph. In this paper, we present a novel method, termed **Spe**ctral **A**pp**r**oach for ZNC (SpeAr), that incorporates spectral analysis and learnable class prototypes to discover the intrinsic cluster structure embedded within the graph. Specifically, we introduce a spectral contrastive loss for optimizing node embedding and posit that minimizing this loss is approximate to conducting a spectral decomposition of the graph. The spectral contrastive loss can elucidate the intrinsic structure of graph data, ensuring a high degree of discriminability among classes in the latent space. In addition, we initiate the class prototypes by class semantic vectors and iteratively refine these prototypes based on the node embeddings.

In summary, our contribution is three-fold: (1) We propose a spectral contrastive loss to update nodes and establish an approximate relationship between the node embeddings obtained from this loss and the feature vectors derived from spectral decomposition. (2) Taking class semantic vectors as the initial class prototypes, we iterative refine these prototypes building upon the node embeddings. (3) Experiments demonstrate that our proposed method, SpeAr, can alleviate the bias issue that occurs in existing ZNC tasks, thereby enhancing the model's ability to recognize unseen classes.

## 2   Related Work

**Zero-shot learning.** Zero-shot learning [14] has gained attention for its potential for a wide range of applications in fields such as image processing [15], speech recognition [16], and natural language processing [17]. In zero-shot learning models, commonly used external semantic knowledge includes attribute vectors and semantic vectors, among others. In particular, text vectors play an important role in representing textual content, and they translate words, sentences, and even whole paragraphs into vector representations through techniques such as Word2Vec [18], GloVe [19], and BERT [20]. Existing methods focus on establishing mapping relationships between samples and external knowledge for effective knowledge transfer to unseen classes. These include direct mapping methods [21, 22], which work by constructing a mapping function between samples and external knowledge and optimizing it; generation-based methods [15, 23], which train semantic-visual generators to generate samples for the unseen classes; and local embedding-based methods [24, 25], which use the attributes to guide the learning and migration of discriminative local embeddings. While the aforementioned methods have exhibited noteworthy progress in managing regular data types like images, they encounter obstacles in handling ZNC. The distinct structural attributes of graph data often hinder traditional zero-shot learning techniques from capturing and addressing these crucial properties, thereby rendering them ineffective in resolving the ZNC issue.

**Node classification.** Node classification represents a pivotal task within the domain of graph data analysis. The introduction of GNNs has catalyzed a transformative evolution in the advancement of

node classification techniques [7]. GNNs possess the capability to concurrently process the adjacency structures and feature information of nodes. A multitude of methods predicated on GNNs [26, 27] have been introduced, progressively emerging as the predominant technologies for addressing node classification challenges. These conventional approaches are largely predicated on the assumption that the class labels present in the training data encompass all possible classes. However, the reality is that new categories continually emerge. Regrettably, existing GNN frameworks exhibit notable deficiencies when it comes to effectively handling unlabeled nodes from unseen classes.

## 3    A Spectral Approach for Zero-shot Node Classification

### 3.1    Problem Formalization

Let $\mathcal{G} = \{\mathcal{V}, \mathcal{E}\}$ denote a **graph**. $\mathcal{V}$ is the set of nodes with $|\mathcal{V}| = N$, $\mathcal{E}$ is the set of edges. In the graph, the feature matrix is $X \in R^{N \times d}$, $d$ is the dimension of features. $A \in \{0,1\}^{N \times N}$ denotes the adjacency matrix, where $a_{ij} = 1$ if $(v_i, v_j) \in \mathcal{E}$, otherwise $a_{ij} = 0$.

For **zero-shot node classification**, $\mathcal{C} = \{c_1, ..., c_{|\mathcal{C}|}\}$ is the class set, $\mathcal{C}_s = \{c_1, ..., c_{|\mathcal{C}_s|}\}$ is the seen class set, $\mathcal{C}_u = \{c_1, ..., c_{|\mathcal{C}_u|}\}$ is the unseen class set, $\mathcal{C}_s \cup \mathcal{C}_u = \mathcal{C}$ and $\mathcal{C}_s \cap \mathcal{C}_u = \phi$. Given an $x \in X$, we use $\mathcal{P}$ to denote the marginal distributions of all data that $x \sim \mathcal{P}$, and let $\mathcal{P}_{c_i}$ denote the distribution of labeled samples with class label $c_i \in \mathcal{C}$ and $\mathcal{P}_u$ denote those of unlabeled data. In addition, every class has a distinct class semantic vector $s_i \in R^{d_1}$, $|\mathcal{C}|$ class semantic vectors (CSVs) (Further details are provided in Section 4.1) can be formed into a matrix $S \in R^{|\mathcal{C}| \times d_1}$. In graph $\mathcal{G}$, there are $N_s$ labeled nodes from seen classes, forming the label matrix $Y_s \in \mathbb{R}^{N_s \times |\mathcal{C}_s|}$. For ZNC, we predict labels for $N_u$ unlabeled nodes from unseen classes, with predictions in $\bar{Y}_u \in \mathbb{R}^{N_u \times |\mathcal{C}_u|}$. Generalized zero-shot node classification (GZNC) predicts labels for $N_{su}$ unlabeled nodes, including both seen and unseen classes, with predictions in $\bar{Y}_{su} \in \mathbb{R}^{N_{su} \times |\mathcal{C}|}$.

For **node embedding**, we employ the GNN [7] model for node representations. Specifically, given the adjacency matrix $A$ and the feature matrix $X$, aggregating information from neighboring nodes, the node embedding is:

$$g(X) = \sigma(D^{-1/2} A D^{-1/2} X W_1), \tag{1}$$

where $D_{ii} = \sum A_i$ is the diagonal matrix of node degrees. $W_1$ denotes the parameters of a single-layer neural network, and $\sigma(\cdot)$ is activation function. In addition to aggregating information about neighboring nodes, this study also emphasizes the importance of information about the nodes themselves in category mining. Therefore, we input each node into the neural network to obtain its representation, the embedding of nodes in the latent space is formulated as follows:

$$f(X) = \sigma(D^{-1/2} A D^{-1/2} X W_1) + \sigma(X W_2), \tag{2}$$

where $W_2$ denotes the parameters of a single-layer neural network.

### 3.2    Preliminaries of Spectral Decomposition

Given a graph structure, we primarily employ spectral decomposition as a key technique to derive principled embeddings. Spectral decomposition, a robust mathematical instrument, has long exhibited its unparalleled value in clustering algorithms. Specifically, the spectral clustering methodology [28, 29, 30] utilizes spectral decomposition [31, 32] to learn the embeddings of samples in a designated space, subsequently enabling efficient cluster analysis through algorithms like $K$-means [28].

The spectral decomposition can be succinctly expressed as $\tilde{A} = Q \Lambda Q^T$, wherein $\tilde{A} = D^{-1/2} A D^{-1/2}$, $Q$ denotes an orthogonal matrix encapsulating the complete set of eigenvectors of $\tilde{A}$. $\Lambda$ signifies a diagonal matrix whose diagonal entries are constituted by the eigenvalues of $\tilde{A}$, corresponding to the eigenvectors in $Q$. Let $\lambda_1, \lambda_2, ..., \lambda_k$ denote the top-$k$ eigenvalues, and $q_1, q_2, ..., q_k$ represent the corresponding top-$k$ eigenvectors. We define $F^* = [q_1, q_2, ..., q_k]^T \in R^{N \times k}$ as the matrix of eigenvector moments, which serves as a novel, condensed representation of the sample. These eigenvectors embody the most significant directions within the data. Let $z_i$ be the $i^{th}$ row of the matrix $F^*$, It turns out that $z_i$ can serve as desirable node embeddings of $x_i$.

### 3.3 Spectral Zero-shot Node Representation Learning

Spectral decomposition, by leveraging the relationships between data points, effectively unearths the implicit class cluster structures within a graph, thereby obtaining node representations with class discriminability. One of the fundamental challenges of ZNC is identifying unlabeled nodes of unseen categories. These unlabeled nodes contain cluster information, and utilizing the information is important for enhancing the model's recognition and understanding of unseen categories. By revealing the latent class cluster structures in the graph through spectral decomposition, we offer a novel perspective for the ZNC.

It is imperative to consider not only the relationships between nodes within the unlabeled set but also the labeled information in the seen classes. On this basis, we reconfigure the adjacency matrix $A$ by integrating the labeled information, so that $A = \alpha A^l + \beta A$, where $A^l$ represents the label relation matrix. We use $A_{xx'}$ to denote the entries of the reshaped $A$, where $A_{xx'} = 0$ represents that $x$ and $x'$ neither belong to the same category nor have an adjacency relationship. Otherwise, $A_{xx'} \neq 0$ indicates that at least one of the two conditions is met.

$$A_{xx'} = \alpha \sum_{c_i \in \mathcal{C}} \mathbb{E}_{x \sim \mathcal{P}_{c_i}, x' \sim \mathcal{P}_{c_i}} A_{xx'}^l + \beta \mathbb{E}_{x \sim \mathcal{P}_u, x' \sim \mathcal{P}_u} A_{xx'}, \tag{3}$$

and thus, we have $A_x = \sum_{x' \in X} A_{xx'}$. In this section, based on the reshaped adjacency matrix $A$ and its normalization $\tilde{A}$, we refer to the Eckart-Young-Mirsky theorem [33], where the loss of solving $F^*$ can be formalized as:

$$\min_{F \in \mathbb{R}^{N \times k}} \mathcal{L}_{sd}(F, \tilde{A}) = ||\tilde{A} - FF^\top||_F. \tag{4}$$

Now, we view $\mathbf{f}_x^\top$ of $F$ as a scaled version of learned feature embedding $f : X \mapsto R^k$. $\mathcal{L}_{sd}$ can be formulated as a variant of the contrastive learning objective, enabling us to theoretically establish the approximation between the learned node representations and the top-$k$ singular vectors of $\tilde{A}$. We formalize the approximation in Theorem 3.1.

**Theorem 3.1.** *We define* $\mathbf{f}_x = \sqrt{A_x} f(x)$ *for some function* $f$, $\alpha, \beta$ *are hyper-parameters. Then minimizing the loss function* $\mathcal{L}_{sd}(F, \tilde{A})$ *is equivalent to minimizing the following loss function for* $f$, *which we term spectral contrastive loss,*

$$\mathcal{L}_{scl}(f) \triangleq -2\alpha \mathcal{L}_1(f) - 2\beta \mathcal{L}_2(f) + \alpha^2 \mathcal{L}_3(f) + 2\alpha\beta \mathcal{L}_4(f) + \beta^2 \mathcal{L}_5(f), \tag{5}$$

*where*

$$\mathcal{L}_1(f) = \sum_{c_i \in \mathcal{C}} \mathbb{E}_{x \sim \mathcal{P}_{c_i}, x^+ \in \{x' | A_{xx'} \neq 0, x' \sim \mathcal{P}_{c_i}\}} f(x)^\top f(x^+),$$

$$\mathcal{L}_2(f) = \mathbb{E}_{x \sim \mathcal{P}_u, x^+ \in \{x' | A_{xx'} \neq 0, x' \sim \mathcal{P}_u\}} f(x)^\top f(x^+),$$

$$\mathcal{L}_3(f) = \sum_{c_i \in \mathcal{C}} \sum_{c_j \in \mathcal{C}} \mathbb{E}_{x \sim \mathcal{P}_{c_i}, x^- \in \{x' | A_{xx'} = 0, x' \sim \mathcal{P}_{c_j}\}} [(f(x)^\top f(x^-))^2],$$

$$\mathcal{L}_4(f) = \sum_{c_i \in \mathcal{C}} \mathbb{E}_{x \sim \mathcal{P}_{c_i}, x^- \in \{x' | A_{xx'} = 0, x' \sim \mathcal{P}_u\}} [(f(x)^\top f(x^-))^2],$$

$$\mathcal{L}_5(f) = \mathbb{E}_{x \sim \mathcal{P}_u, x^- \in \{x' | A_{xx'} = 0, x' \sim \mathcal{P}_u\}} [(f(x)^\top f(x^-))^2].$$

*Proof.* (sketch) We can expand $\mathcal{L}_{sd}(F, A)$ and obtain

$$\mathcal{L}_{sd}(F, A) = \sum_{x, x' \in X} \left( \frac{A_{xx'}}{\sqrt{A_x A_{x'}}} - \mathbf{f}_x^\top \mathbf{f}_{x-} \right)^2$$

$$= \text{const} + \sum_{x, x' \in X} \left( -2A_{xx'} f(x)^\top f(x') + A_x A_{x'} (f(x)^\top f(x'))^2 \right). \tag{6}$$

The first term is a constant. The form of $\mathcal{L}_{scl}(f)$ is derived from plugging $A_{xx'}$ and $A_x$. □

**Analysis of Theorem 3.1.** The $\mathcal{L}_1$ loss is tailored to reduce the distances between embeddings of label nodes sharing identical class labels, thereby enhancing intra-class compactness. The $\mathcal{L}_2$ loss targets the unlabeled node pair with the highest adjacency probability as a positive pair, amplifying the discriminative capacity of the embedding space. Conversely, the $\mathcal{L}_3$, $\mathcal{L}_4$, and $\mathcal{L}_5$ losses are dedicated to the strategic dispersion of embeddings associated with negative pairs. The $\mathcal{L}_3$ loss function is calibrated to induce a distinct separation between embeddings of labeled nodes with different class labels. The $\mathcal{L}_4$ loss treats labeled nodes and unlabeled nodes counterparts as negative pairs. Finally, the $\mathcal{L}_5$ loss further refines the embedding space by considering all remaining unlabeled node pairs, apart from those identified as positive in the $\mathcal{L}_2$ loss, as negative pairs. Above all, these intricately designed loss $\mathcal{L}_{scl}(f)$ facilitates the aggregation of similar nodes and the segregation of dissimilar ones within the embedding space significantly enhancing the representational efficacy of the embeddings.

The Theorem 3.1 of this section is primarily adapted from Theorem 4.1 in [29] and Theorem 3.1 in [34]. The main differences from the original theorems are reflected in the following two aspects: First, the types of data objects processed are different; the proposed method focuses on graph data objects that inherently possess structural characteristics. Second, during the construction of the loss function, we define an unlabeled node pair exhibiting the highest adjacency probability as a positive pair. These adjustments make the Theorem 3.1 more aligned with the specific context and requirements of the research presented in this paper.

## 3.4 Class Prototype Learning

The proposed method employs a meticulously crafted spectral contrastive loss function to ensure the inter-class separability of the embedding vectors obtained in the latent space. To achieve classification of unseen class nodes, we utilize valuable pseudo-label information to perform iterative updates on class prototypes, which serve as the centers for each class. These prototypes are initially instantiated with semantic vectors $S$ that are rich in categorical information. Utilizing these vectors, we perform preliminary classification predictions for the nodes, thereby creating pseudo-labels. Based on these pseudo-labels, the unlabeled nodes with pseudo-label predictions exceeding a preset threshold $q$ are selscted to update unseen class prototypes. For labeled nodes with pseudo-label predictions matching the ground-truth label, we use them to refine prototypes for seen classes. Our update rule is defined as:

$$p_c = \begin{cases} (1 - \mu) \cdot p_c + \mu \cdot z & \text{if } \bar{y} = c \text{ and } \Pr(\bar{y}) > q, \bar{y} \text{ is the pseudo-label of } z \\ p_c & \text{otherwise.} \end{cases} \tag{7}$$

Here, we use $\Pr(\bar{y}) > q$ to denote the selected nodes whose pseudo-label predicted probability exceeds threshold $q$. $\mu$ is the updated parameter. This strategy not only significantly enhances the model's generalization capability for unseen classes but also ensures that the classification accuracy for seen classes is maintained and enhanced.

## 3.5 Training and Testing

Considering the significant modal differences between the initial class prototypes (i.e., semantic vectors) and node embedding, we design a carefully planned two-stage model training approach. This strategy aims to gradually narrow the inter-modal gap through a careful optimization process, leading to improved and refined class prototypes. In the first phase, we focus on roughly tuning the prototype to establish a solid starting point. Subsequently, in the second phase, labeling and adjacency-guided refinement of the prototypes ensures the quality of prototype learning. Through this staged training, our model can achieve a more accurate representation of the class prototypes.

**In the first phase**, the backbone is pre-trained using unsupervised spectral contrastive loss $\mathcal{L}_{uscl}$. $\mathcal{L}_{uscl}$ means that the positive samples are the nodes themselves, and the negative samples are selected from other nodes in the graph. **In the second phase**, the model is trained using $\mathcal{L}_{scl}$.

**During the test phase**, the embedding $z_i$ of $x_i$ in the latent space is obtained by the network. Based on the learned class prototypes, we predict its label by :

$$c^* = \arg \max_{c \in \mathcal{C}_u/\mathcal{C}} (z \times p_c), \tag{8}$$

$c \in \mathcal{C}_u/\mathcal{C}$ corresponds to ZNC/GZNC tasks.

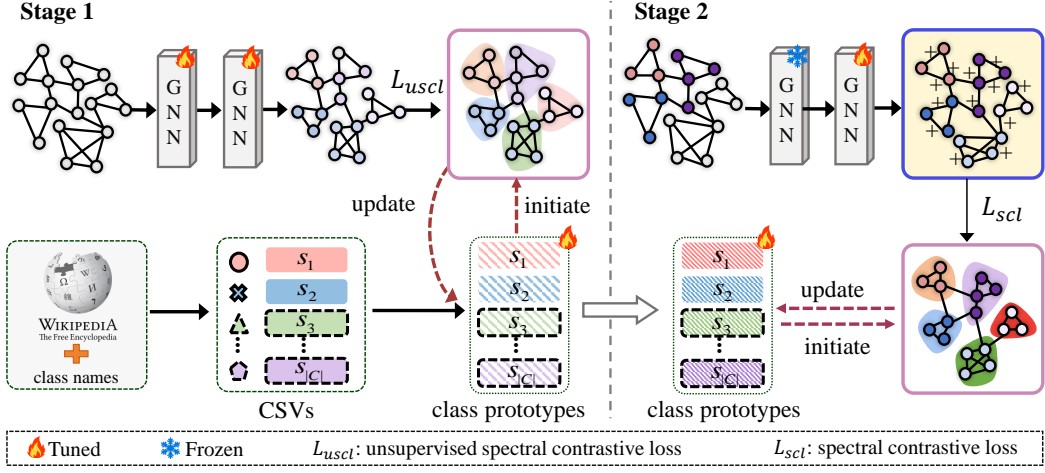

Figure 1: The overall framework for SpeAr. The whole training process consists of two stages.

## 4 Experiment

### 4.1 Experimental Setup

**Datasets.** Following the methods DGPN [10], DBiGCN [12], and GraphCEN [13], we seek to substantiate the validity of our proposed SpeAr through its application to three public datasets: Cora [1], Citeseer [1], C-M10M [35]. To ensure equitable comparison, the data partitioning strategy mirrors that of the aforementioned methods. The dataset details are shown in Table 1. For Class Split I and II, the evaluation criterion is classification accuracy. In addition to that, we also give the Class Split III to validate the effectiveness of SpeAr in handling GZNC. The evaluation criterion is $H$, defined as $H = 2 \times (seen \times unseen)/(seen + unseen)$. The $seen$ and $unseen$ are the classification accuracies of seen and unseen classes, respectively.

For class semantic vectors (CSVs), DGPN has delineated two primary categories: label-based CSVs, which are word embeddings derived from class names, and text-based CSVs, which are document embeddings textual descriptions related to the class. These vectors are extracted utilizing the esteemed natural language processing model, Bert-Tiny [36]. In our experiments, we mainly use text-based CSVs because text contains richer information. Furthermore, we examine the different impacts of employing distinct CSVs on the experimental outcomes.

**Baselines Methods.** DGPN draws upon a suite of comparative methods rooted in traditional zero-shot learning. We follow DGPN and list algorithms such as DAP [37], ESZSL [21], ZS-GCN [38], WDVSc [39], Hyperbolic-ZSL [40], DAP and ZS-GCN, as well as their CNN-enhanced counterparts, DAP(CNN) and ZS-GCN(CNN). These conventional methods, predominantly tailored for visual domain data (e.g., image data) prediction are often ill-equipped to handle the graph data, thereby exhibiting limitations in effectively addressing the ZNC problem. Our study places a particular emphasis on evaluating the performance of our proposed SpeAr model against these established methods such as DGPN [10], DBiGCN [12], GraphCEN [13], underscoring the superior effectiveness of SpeAr in the context of ZNC.

**Parameter Settings.** In SpeAr, we employ a two-stage training strategy. During the first phase, the model is trained by the loss $\mathcal{L}_{uscl}$, which is governed by parameters $\alpha$ and $\beta$. In this stage, samples are utilized for updating category prototypes only if the predicted probability of pseudo-labels exceeds a predefined threshold $q$, with $q$ restricted to the range $\{0.5, 0.6, 0.7, 0.8, 0.9\}$. To ensure the stability of the update process, the prototype update parameter $\mu$ is cautiously set to 0.1 to prevent the introduction of erroneous information that might arise from higher values. In the second phase, the loss function $\mathcal{L}_{scl}$ continues with the parameters $\alpha$ and $\beta$ established in the first phase. Here, we refine the prototype update process by selecting the top-$s$ nodes with the highest pseudo-label probabilities, with $s$ uniformly set to 1000 across all datasets. Furthermore, to achieve fine-tuning of the prototypes, the update parameter $\mu$ is further reduced to 0.01 in this phase.

Table 1: Summary of the datasets.

| Dataset | Nodes | Edges | Features | Classes | ZNC | | GZNC |
| | | | | | Class Split I [Train/Val/Test] | Class Split II [Train/Val/Test] | Class Split III [Train/Val/Test] |
|---------|-------|-------|----------|---------|------|------|------|
| Cora | 2708 | 5429 | 1433 | 7 | [3/0/4] | [2/2/3] | [3/0/7] |
| Citeseer | 3327 | 4732 | 3703 | 6 | [2/0/4] | [2/2/3] | [2/0/6] |
| C-M10M | 4464 | 5804 | 128 | 6 | [3/0/3] | [2/2/3] | [3/0/6] |

## 4.2 Experimental Results

Table 2 records the ZNC accuracies corresponding to Class Split I and II. A thorough analysis of Table 2 yields the following insights:

a) For Class Split I, SpeAr achieves a notable accuracy enhancement on the Cora, Citeseer, and C-M10M datasets when compared to existing methods. This enhancement underscores the potency of SpeAr's spectral decomposition and prototype updating tactics. By delving deeper into the intrinsic class cluster structures embedded within unlabeled data, SpeAr significantly amplifies its comprehension and identification of unseen classes, while ensuring good separability between different classes.

(b) For Class Split II, SpeAr shows a significant advantage over the state-of-the-art methods. In this scenario, model parameters are determined using the validation set. This approach is justified within the methodology of our study, as our objective is to thoroughly explore the information across all classes. We believe that a model optimized using validation sets can achieve a comprehensive understanding of the classes and is applicable to some extent to the recognition of unseen classes.

Table 2: Zero-shot node classification accuracy (%).

| | | Cora | Citeseer | C-M10M |
|---|---|------|----------|--------|
| **Class Split I** | RandomGuess | 25.35 | 24.86 | 33.21 |
| | DAP | 26.56 | 34.01 | 38.71 |
| | DAP (CNN) | 27.80 | 30.45 | 32.97 |
| | ESZSL | 27.35 | 30.32 | 37.00 |
| | ZS-GCN | 25.73 | 28.62 | 37.89 |
| | ZS-GCN (CNN) | 16.01 | 21.18 | 36.44 |
| | WDVSc | 30.62 | 23.46 | 38.12 |
| | Hyperbolic-ZSL | 26.36 | 34.18 | 35.80 |
| | DGPN | 33.78 | 38.02 | 41.98 |
| | DBiGCN | 45.14 | 40.97 | 45.45 |
| | GraphCEN | 48.43 | 40.77 | 44.17 |
| | SpeAr (Ours) | **60.48** | **59.72** | **54.22** |
| | Improve | 24.88% | 45.77% | 19.30% |
| **Class Split II** | RandomGuess | 32.69 | 50.48 | 49.73 |
| | DAP | 30.22 | 53.30 | 46.79 |
| | DAP (CNN) | 29.83 | 50.07 | 46.29 |
| | ESZSL | 38.82 | 55.32 | 56.07 |
| | ZS-GCN | 29.53 | 52.22 | 55.28 |
| | ZS-GCN (CNN) | 33.20 | 49.27 | 51.37 |
| | WDVSc | 34.13 | 52.70 | 46.26 |
| | Hyperbolic-ZSL | 37.02 | 46.27 | 55.07 |
| | DGPN | 46.40 | 61.90 | 62.46 |
| | DBiGCN | 49.20 | 60.11 | 71.86 |
| | GraphCEN | 50.61 | 60.47 | 70.83 |
| | SpeAr (Ours) | **58.20** | **75.13** | **79.00** |
| | Improve | 15.00% | 21.37% | 9.94% |

Table 3 records the GZNC accuracies corresponding to Class Splits III. A thorough analysis of Table 3 yields the following insights: We record the performance of the DGPN and DBiGCN in addressing the GZNC problem. Table 3 shows that both DGPN and DBiGCN exhibit suboptimal performance when tackling GZNC. The classification accuracies for unseen classes are less than satisfactory and there is a significant prediction bias. In contrast, SpeAr concurrently learns the prototypes for both seen and unseen classes during its training process. Experimental results indicate that SpeAr achieves a significant enhancement in classification accuracy of unseen classes compared to DGPN and DBiGCN, and gained a better $H$ value. This enhancement indicates that SpeAr bolsters the model's comprehension of unseen classes through more intensive mining of the clustering information embedded within unlabeled nodes, thereby enhancing inter-class separability and partially mitigating prediction bias. However, we also note that despite the progress made by SpeAr, there is still room for improving its performance.

Table 3: Generalized zero-shot node classification accuracy (%).

| | Cora | | | Citeseer | | | C-M10M | | |
|---|---|---|---|---|---|---|---|---|---|
| | *seen* | *unseen* | $H$ | *seen* | *unseen* | $H$ | *seen* | *unseen* | $H$ |
| DGPN | **94.26** | 0 | 0 | **89.76** | 0 | 0 | **96.71** | 0 | 0 |
| DBiGCN | 66.39 | 1.27 | 2.50 | 35.70 | 4.83 | 8.50 | 30.04 | 4.30 | 7.52 |
| SpeAr(Ours) | 34.84 | **26.32** | **29.98** | 27.17 | **25.78** | **26.45** | 26.75 | **18.95** | **22.19** |

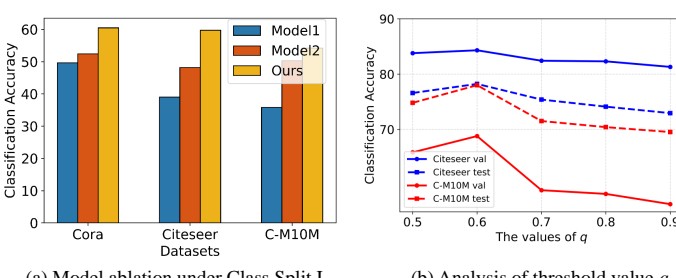

(a) Model ablation under Class Split I

(b) Analysis of threshold value $q$

Figure 2: Ablation study and parametric analysis of threshold value $q$.

Future work will further explore how to optimize the model to achieve superior GZNC classification results.

### 4.3 Ablation Study

In SpeAr, the loss $L_{scl}$ plays an essential role, as the node embeddings optimized by it approximate the eigenvectors obtained through spectral decomposition. By optimizing the $L_{scl}$, we not only capture the local connectivity information of the graph but also uncover the implicit class cluster structures within the unlabeled nodes, thereby achieving a comprehensive of the class information depicted in the graph. In Figure 2 (a), the Model1 indicates that upon the removal of the $L_{scl}$ loss from SpeAr, its performance experiences a significant degradation, especially on the Citeseer dataset. This result strongly demonstrates that the $L_{scl}$, by effectively leveraging the relationships between nodes in the graph, deeply mines the categorical information on the graph and enhances the discriminability between classes.

Furthermore, the mechanism of prototype updating is equally crucial. Random initialization of prototypes can lead to instability in the results. Therefore, this study employs semantic vectors as the initial values for prototypes and devises a two-stage training strategy. In the first phase, we utilize an unsupervised spectral contrastive loss for node embedding. In the second phase, we take the updated prototypes from the first phase as the initial values and further optimize the representations of nodes and prototypes using a supervised spectral contrastive loss. In Figure 2 (a), Model2 is the model that omits the first phase and directly uses semantic vectors as the initial prototypes for the second phase. The results decline across three datasets. This phenomenon further confirms the effectiveness of our designed two-stage training strategy.

### 4.4 Parametric Analysis

The threshold value $q$ is instrumental in determining the subset of nodes that contribute to the prototype update process. When a sample's pseudo-label probability exceeds the threshold $q$, it updates the class prototypes. The choice of $q$ depends on model and dataset: low $q$ risks noisy pseudo-labels, degrading performance, high $q$ ensures accuracy but limits unlabeled data utilization. In SpeAr, select $q$ using the validation set. As shown in Figure 2 (b), across three datasets, as $q$ varies, the accuracy changes in both the test and validation sets are roughly consistent, confirming the strategy's effectiveness and reliability.

In SpeAr, $L_{scl}$ includes two pivotal parameters $\alpha$ and $\beta$, deciding the relative significance of labeled and unlabeled samples, respectively. As illustrated in the Figure 3, the selection range for $\alpha$ and $\beta$ is constrained to the set $\{0.25, 0.5, 0.75, 1\}$. We record the effects of $\alpha$ and $\beta$ on Cora, Citeseer, and C-M10M. The $\alpha$ and $\beta$ regulate the contribution of labeled and unlabeled samples in overall loss. The

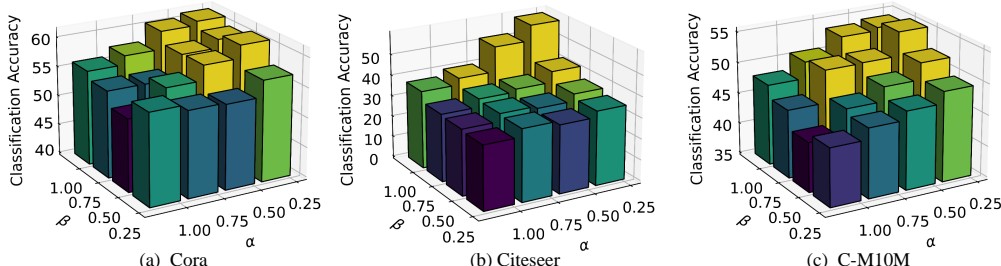

Figure 3: The effects of $\alpha$ and $\beta$ on Cora, Citeseer, and C-M10M.

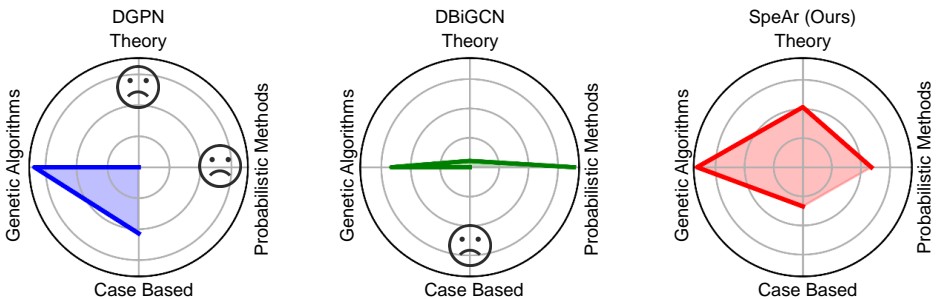

Figure 4: An example of the SpeAr's effectiveness in mitigating prediction bias on Cora dataset.

results reveal a notable trend: when the value of $\beta$ is larger than $\alpha$, the model generally exhibits better performance, with this advantage being particularly prominent when $\beta = 1$. The conclusions meet our expectations. A larger $\beta$ can reduce overfitting to seen classes and better exploit the clustering information embedded in unlabeled nodes. Thus, the parameters need to satisfy $\beta > \alpha$. This suggests that the $\alpha$ and $\beta$ are not sensitive to some extent.

### 4.5 Further Analysis.

Table 3 demonstrates the significant effectiveness of the SpeAr model in mitigating predictive bias, with marked improvements in classification accuracy for both seen and unseen classes. In addition, for Cora, Citeseer, and C-M10M, we find that existing models fail to recognize some of the categories. Initially, we define recall as the proportion of samples correctly classified into a class relative to the total number of samples in that class. As shown in Figure 4 , the recall of certain classes is extremely low or even zero for DGPN and DBiGCN on Cora dataset. For instance, the unseen classes "Probabilistic Methods" and "Theory" of Cora exhibit a zero recall rate when predicted by the DGPN model. This reveals the prediction bias issue inherent in current methods when addressing the ZNC problem, where some classes are not correctly identified, and unseen class nodes are erroneously concentrated in a few incorrect classes. In contrast, the SpeAr model provides more accurate classification outcomes for all unseen classes. These results further confirm that the SpeAr model can improve the recognition of unseen classes and enhance class separability by utilizing and mining the class cluster information in unlabeled nodes.

SpeAr has demonstrated exceptional performance in addressing the ZNC problem. We argue that this methodology is equally efficacious when applied to conventional zero-shot learning endeavors, which are predominantly concerned with the identification of unseen classes within the Euclidean space. When applying SpeAr to such tasks, the central challenge lies in effectively constructing the adjacency relationships between samples and integrating them with the loss function proposed in this paper.

## 5   Conclusion

This paper introduces a spectral method designed to address the challenge of ZSNC, ensuring the effective unearthing of class cluster structures within graphs and enhancing the separability

between classes. Unlike existing approaches that focus on leveraging external knowledge to mitigate predictive bias, our novel approach SpeAr accentuates the exploration of the inherent, implicit cluster information encapsulated within the data of unlabeled nodes. SpeAr optimizes node embeddings by minimizing spectral contrast loss and iteratively updates the class prototypes with semantic vectors as initialization. Empirical results demonstrate that SpeAr achieves significant accuracy improvements in tackling ZNC and GZNC problems, effectively alleviating predictive bias. In future research, we plan to extend the concepts of this study to other tasks with similar training data, thereby further validating its universality and efficacy.

## Acknowledgement

Research is supported by the National Science and Technology Major Project (2020AAA0106102), the National Natural Science Foundation of China (62376141), and the Natural Science Foundation of Shanxi Province, China (202203021222075). Any opinions, findings, conclusions, or recommendations expressed in this material are those of the authors and do not necessarily reflect the views, policies, or endorsements either expressed or implied, of the sponsors.

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

# 6 Appendix material

## 6.1 Proof of Theorem 3.1

*Proof.* We can expand $\mathcal{L}_{sd}(F, A)$ and obtain

$$\mathcal{L}_{sd}(F, A) = \sum_{x,x' \in X} \left( \frac{A_{xx'}}{\sqrt{A_x A_{x'}}} - \mathbf{f}_x^\top \mathbf{f}_{x'} \right)^2$$
$$= \text{const} + \sum_{x,x' \in X} \left( -2 A_{xx'} f(x)^\top f(x') + A_x A_{x'} \left( f(x)^\top f(x') \right)^2 \right), \qquad (9)$$

where $\mathbf{f}_x = \sqrt{A_x} f(x)$ is a re-scaled version of $f(x)$. At a high level, we follow the proof in [30], while the specific form of loss varies with the different definitions of positive/negative pairs. The form of $\mathcal{L}_{scl}(f)$ is derived from plugging $A_{xx'}$ and $A_x$.

Recall that $A_{xx'}$ is defined by

$$A_{xx'} = \alpha \sum_{c_i \in \mathcal{C}} \mathbb{E}_{x \sim \mathcal{P}_{c_i}, x' \sim \mathcal{P}_{c_i}} A_{xx'}^l + \beta \mathbb{E}_{x \sim \mathcal{P}_u, x' \sim \mathcal{P}_u} A_{xx'}, \qquad (10)$$

thus we have,

$$-2 \sum_{x,x' \in X} A_{xx'} f(x)^\top f(x')$$
$$= -2\alpha \sum_{c_i \in \mathcal{C}} \mathbb{E}_{x \sim \mathcal{P}_{c_i}, x' \sim \mathcal{P}_{c_i}} A_{xx'} f(x)^\top f(x') - 2\beta \mathbb{E}_{x \sim \mathcal{P}_u, x' \sim \mathcal{P}_u} A_{xx'} f(x)^\top f(x')$$
$$= -2\alpha \sum_{c_i \in \mathcal{C}} \mathbb{E}_{x \sim \mathcal{P}_{c_i}, x' \sim \mathcal{P}_{c_i}, A_{xx'} \neq 0} f(x)^\top f(x') - 2\beta \mathbb{E}_{x \sim \mathcal{P}_u, x' \sim \mathcal{P}_u, A_{xx'} \neq 0} f(x)^\top f(x')$$
$$= -2\alpha \sum_{c_i \in \mathcal{C}} \mathbb{E}_{x \sim \mathcal{P}_{c_i}, x^+ \in \{x' | A_{xx'} \neq 0, x' \sim \mathcal{P}_{c_i}\}} f(x)^\top f(x^+) - 2\beta \mathbb{E}_{x \sim \mathcal{P}_u, x^+ \in \{x' | A_{xx'} \neq 0, x' \sim \mathcal{P}_u\}} f(x)^\top f(x^+)$$
$$= -2\alpha \mathcal{L}_1(f) - 2\beta \mathcal{L}_2(f).$$

The penultimate equation is derived from the following lemma:

$$\mathcal{L}_{sd}(F) = \mathcal{L}(f) + \text{const}$$
$$\text{where } \mathcal{L}(f) \triangleq -2 \cdot \mathbb{E}_{x,x^+} \left[ f(x)^\top f(x^+) \right] + \mathbb{E}_{x,x^-} \left[ \left( f(x)^\top f(x^-) \right)^2 \right]. \qquad (11)$$

Recall that $A_x$ is given by

$$A_x = \sum_{x' \in X} A_{xx'} \qquad (12)$$
$$= \alpha \sum_{c_i \in \mathcal{C}} \mathbb{E}_{x \sim \mathcal{P}_{c_i}} A_x + \beta \mathbb{E}_{x \sim \mathcal{P}_u} A_x. \qquad (13)$$

thus plugging $A_x$ and $A_{x'}$ we have,

$$\sum_{x,x' \in X} A_x A_{x'} \left( f(x)^\top f(x') \right)^2$$

$$= \sum_{x,x' \in X} \left( \alpha \sum_{c_i \in \mathcal{C}} \mathbb{E}_{x \sim \mathcal{P}_{c_i}} A_x + \beta \mathbb{E}_{x \sim \mathcal{P}_u} A_x \right) \cdot \left( \alpha \sum_{c_j \in \mathcal{C}} \mathbb{E}_{x' \sim \mathcal{P}_{c_j}} A_{x'} + \beta \mathbb{E}_{x' \sim \mathcal{P}_u} A_{x'} \right) \left( f(x)^\top f(x') \right)^2$$

$$= \alpha^2 \sum_{x,x' \in X} \sum_{c_i \in \mathcal{C}} \mathbb{E}_{x \sim \mathcal{P}_{c_i}} A_x \sum_{c_j \in \mathcal{C}} \mathbb{E}_{x' \sim \mathcal{P}_{c_j}} A_{x'} \left( f(x)^\top f(x') \right)^2$$

$$+ 2\alpha\beta \sum_{x,x' \in X} \sum_{c_i \in \mathcal{C}} \mathbb{E}_{x \sim \mathcal{P}_{c_i}} A_x \mathbb{E}_{x' \sim \mathcal{P}_u} A_{x'} \left( f(x)^\top f(x') \right)^2$$

$$+ \beta^2 \sum_{x,x' \in X} \mathbb{E}_{x \sim \mathcal{P}_u} A_x \mathbb{E}_{x' \sim \mathcal{P}_u} A_{x'} \left( f(x)^\top f(x') \right)^2$$

$$= \alpha^2 \sum_{c_i \in \mathcal{C}} \sum_{c_j \in \mathcal{C}} \mathbb{E}_{x \sim \mathcal{P}_{c_i}, x^- \in \{x' | A_{xx'}=0, x' \sim \mathcal{P}_{c_j}\}} [\left( f(x)^\top f(x') \right)^2]$$

$$+ 2\alpha\beta \sum_{c_i \in \mathcal{C}} \mathbb{E}_{x \sim \mathcal{P}_{c_i}, x^- \in \{x' | A_{xx'}=0, x' \sim \mathcal{P}_u\}} [\left( f(x)^\top f(x') \right)^2]$$

$$+ \beta^2 \mathbb{E}_{x \sim \mathcal{P}_u, x^- \in \{x' | A_{xx'}=0, x' \sim \mathcal{P}_u\}} [\left( f(x)^\top f(x') \right)^2]$$

$$= \alpha^2 \mathcal{L}_3(f) + 2\alpha\beta \mathcal{L}_4(f) + \beta^2 \mathcal{L}_5(f).$$

$\square$

The proof of Theorem 3.1 is finished.

## 6.2 Additional Experiments

### 6.2.1 Zero-shot node classification for large-scale data

For SpeAr, the spectral contrastive loss computes the similarities between samples, with a time complexity of $O(N_s N_s^+ + N_u N_u^+ + N_s N_u + N_s N_s^- + N_u N_u^-)$. Let $N_s$ be the count of labeled nodes, $N_s^+$ be the count of positive nodes of labeled nodes, and $N_s^-$ the negative nodes. $N_u$ is the count of unlabeled nodes, with $N_u^+$ and $N_u^-$ representing the count of positive and negative nodes, respectively. This complexity reveals a substantial demand for computational resources, presenting a notable challenge for processing large-scale graph data.

Following GraphCEN [13], we validate the efficacy of the SpeAr on large-scale dataset, such as ogbn-arxiv [41]. Ogbn-arxiv has 169343 nodes, and 2484941 edges. The feature dimension is 128 and the total class number is 40. Class split I is [20/0/20], 20 seen classes as training set, 20 unseen classes as testing set. Class split II is [13/13/14], 13 seen classes as training set, 13 unseen classes as validation set, and 14 unseen classes as testing set. Confronted with memory limitations, we adopt a multi-round subgraph extraction strategy. Specifically, in each round, we extract subgraphs that encompass both seen and unseen class nodes and execute the SpeAr algorithm on these subgraphs. Through this iterative process of extraction, we aim to progressively accumulate performance gains that mirror the execution of SpeAr on the entire graph, all while maintaining computational efficiency. As shown in Table 4, our proposed method SpeAr shows significant improvement in performance metrics compared to existing methods. The comparative analysis in the table highlights the superiority of our method in capturing class-discriminative information in graph structures.

Table 4: A comparative performance analysis of DGPN, DBiGCN, and ours SpeAr for zero-shot node classification on ogbn-arxiv. (%)

|  | DGPN | DBiGCN | GraphCEN | SpeAr(Ours) |
|---|---|---|---|---|
| Class Split I | 22.37 | 21.40 | 23.96 | **30.45** |
| Class Split II | 21.95 | 25.92 | 28.36 | **32.20** |

Table 5: The Comparison of zero-shot node classification accuracy (%) using the different CSDs.

| | Cora | | | Citeseer | | | C-M10M | | |
|---|---|---|---|---|---|---|---|---|---|
| | TEXT | LABEL | Decline rate | TEXT | LABEL | Decline rate | TEXT | LABEL | Decline rate |
| DAP | 26.56 | 25.34 | -4.59 % | 34.01 | 30.01 | -11.76% | 38.71 | 32.67 | -15.60% |
| ESZSL | 27.35 | 25.79 | -5.70% | 30.32 | 28.52 | -5.94% | 37.00 | 35.02 | -5.35% |
| ZS-GCN | 25.73 | 23.73 | -7.77% | 28.62 | 26.11 | -8.77% | 37.89 | 33.32 | -12.06% |
| WDVSc | 30.62 | 18.73 | -38.83% | 23.46 | 19.70 | -16.02% | 38.12 | 30.82 | -19.15% |
| Hyperbolic-ZSL | 26.36 | 25.47 | -3.38% | 34.18 | 21.04 | -38.44% | 35.80 | 34.49 | -3.66% |
| DGPN | 33.78 | 32.55 | -3.64% | 38.02 | 31.83 | -16.28% | 41.98 | 35.05 | -16.51% |
| DBiGCN | 45.14 | 39.05 | -13.49% | 40.97 | 39.10 | -3.10% | 45.45 | 43.71 | -3.83% |
| SpeAr(Ours) | **60.48** | **49.52** | -18.12% | **59.72** | **48.88** | -18.15% | **54.22** | **47.05** | -13.22% |

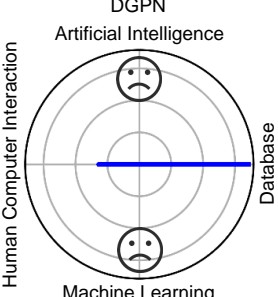 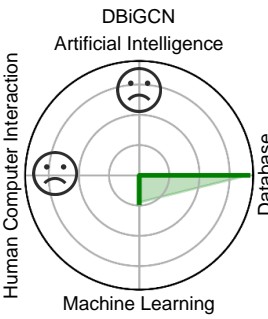 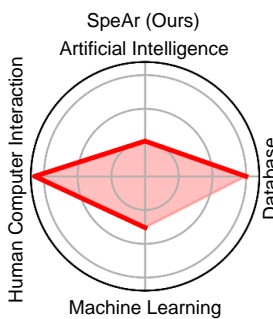

Figure 5: An example of the SpeAr model's effectiveness in mitigating prediction bias on Citeseer.

### 6.2.2 Discussion on Different CSVs

The impact of external knowledge from different sources on model outcomes is significantly varied. In Table 5, we individually examined the effects of LABEL-based CSVs and TEXT-based CSVs as external knowledge. Given that TEXT data encapsulates a richer set of categorical information, the SpeAr model utilizing text-based CSVs demonstrates superior performance. Indeed, when employing LABEL-based CSVs as the input external knowledge, SpeAr also outperforms existing methods, further corroborating the efficacy of the spectral contrastive loss and prototype updating mechanisms proposed in this paper for excavating and identifying categories on graphs. This series of results underscore that our approach significantly enhances the discriminability between different classes, thereby elevating the model's overall recognition capability.

### 6.2.3 Discussion on SpeAr model's effectiveness in mitigating prediction bias

We verify the benefits of SpeAr in mitigating prediction bias on the dataset Citeseer. As shown in Figure 5, the recall for certain classes is extremely low or even zero. For instance, the unseen classes "Human Computer Interaction" and "Artificial Intelligence" exhibit a zero recall rate when predicted by the DBiGCN. In contrast, the SpeAr model provides more accurate classification outcomes for all unseen classes.

## 7 Limitation

Although the SpeAr model shows excellent performance on the ZNC task, its relatively high computational complexity may become a challenge when dealing with large-scale graph data. Especially in application scenarios with limited resources or high real-time requirements, the high computational cost may limit the usefulness of the model. Therefor, we effectively alleviate this problem by adopting the strategy of multi-round subgraph training. The model can gradually learn and integrate information from different subgraphs, thus realizing effective processing of large graph data while maintaining computational efficiency.

## 8 Experiments Compute Resources

Computation resources: We execute our code on a computer with NVIDIA GeForce RTX 3090 (GPU) and Intel Xeon Gold 6254 (CPU).

## 9 Societal Impacts

The introduction of SpeAr has made a significant contribution to the advancement of zero-shot node classification tasks. It demonstrates tremendous potential in the field of data analysis, aiding researchers in uncovering new insights and knowledge. There are no negative societal impacts on our work.

