# OpenReview forum: "SpeAr: A Spectral Approach for Zero-Shot Node Classification"
_NeurIPS.cc/2024/Conference — NeurIPS 2024 poster_

### Official Review · Reviewer_7tvx · 2024-07-09

**Soundness:** 3
**Presentation:** 4
**Contribution:** 3
**Rating:** 7
**Confidence:** 4

**Summary:**

The manuscript proposes the SpeAr method, which leverages spectral analysis and class prototypes to uncover the implicit clustering structures within graphs, providing a comprehensive understanding of node categories. The proposed method establishes an approximate relationship between spectral contrastive loss and spectral decomposition, optimizing node representations by minimizing the loss and iteratively updating class prototypes. Experimental results demonstrate that SpeAr can effectively mitigate the bias issue.

**Strengths:**

1. This paper combines the spectral contrastive learning method with learnable class prototypes to discover implicit cluster information, thereby alleviating the problem of zero-shot node classification prediction bias. It also has relevant theoretical arguments, a novel perspective, and relatively complete theories and experiments.
2. The two-stage training method in this paper plays a key role in improving the representation quality of the prototype and can further improve the performance of zero-shot node classification.
3. Employing spectral clustering and the ingenious design of a spectral contrast loss has efficiently addressed the zero-shot node learning challenge, thereby bridging the gap in current research within this field.

**Weaknesses:**

See the questions listed below.

**Questions:**

1. In Subsection 3.2, what is the choice of k? Regarding the SpeAr algorithm's ability to distinguish between classes, is this strongly correlated with the value of k? Specifically, is there a tendency for the algorithm's ability to discriminate to increase as the value of k increases? We look forward to deeper understanding and insights into this issue.
2. The method part of the paper mentions unsupervised spectral contrast loss, but the corresponding formula is not specifically reflected in the paper. Perhaps the formula is more intuitive.
3. The paper has a very good theoretical analysis. Why spectral contrast loss works for your results.

In conclusion, zero-shot node classification is a problem well worth studying, and this paper introduces a spectral method designed to alleviate the prediction bias problem of zero-shot node classification. This method has achieved good experimental results for zero-shot node classification. In addition, this paper has relevant theoretical arguments, a novel perspective, and relatively complete theories and experiments.

**Limitations:**

Similar to questions.

---

> ### Author Rebuttal · Authors · 2024-08-06
>
> We thank the reviewer for taking the time to review our manuscript and for the valuable comments. Below is a point-by-point response to the comments.
>
> **> Response to Q1: Discussion on $k$**
>
> In Section 3.2 of the main body, the eigenvector matrix is obtained by spectral decomposition, $F^{\ast} = {[q_1, q_2, ..., q_k]}^T \in R^{N\times k}$, which serves as a novel representation of the sample. Let $z_i$ be the $i^{th}$ row of the matrix $F^{\ast}$, It turns out that $z_i$ can serve as desirable node embeddings of $x_i$. Thus, the dimension of the feature space is $k$.
>
> The classification capability of the SpeAr algorithm is closely related to the embedding dimension $k$. Specifically, the choice of $k$ is influenced by the rank of the adjacency matrix. A larger $k$ allows the algorithm to better capture subtle differences between similar samples. In our experiments with three datasets, we chose $k$=2048 to represent a broad range of variations in the data, not just category distinctions.
>
> **> Response to Q2: The unsupervised spectral contrast loss**
>
> Unsupervised contrastive loss pre-trains the model by leveraging the inherent information within the samples. For more information on the importance of pre-training, please refer to the details in **General Response**. Thank you for your patience.
>
> When constructing the unsupervised contrastive loss, each node's positive sample is defined as itself, while the negative samples are all other samples. This approach allows the model to fully utilize the intrinsic information of the data during the pre-training phase, providing more accurate feature representations for subsequent tasks.
>
> **> Response to Q3: The relationship between theory and practical performance**
>
> Given that your concerns align with Reviewer HC2X, we have provided a comprehensive elaboration on the relationship between theory and performance in General Response. We sincerely invite you to review our response in the **General Response**, which we believe will address your questions. We are deeply grateful for your patience and attentive consideration.

---

> > ### Author Response · Authors · 2024-08-11
> >
> > Thanks very much for your thorough review and insightful comments. We hope that our additional evaluations and rebuttal have addressed your primary concerns with our paper. We would really appreciate feedback as to whether there are any (existing or new) points we have not covered, and we would be happy to address/discuss them! Your comments are invaluable in helping us to refine and strengthen our work.

---

> > ### Comment · Reviewer_7tvx · 2024-08-14
> >
> > Thank you for your response. I have thoroughly read the comments from other reviewers and the author's replies, and my concerns have been addressed.

---

### Official Review · Reviewer_HC2X · 2024-07-11

**Soundness:** 3
**Presentation:** 3
**Contribution:** 3
**Rating:** 7
**Confidence:** 4

**Summary:**

Zero-shot node classification is a vital task in the field of graph data processing. Prediction bias is one of the primary challenges in zero-shot node classification. This paper employs spectral analysis coupled with learnable class prototypes to discover the implicit cluster structures within the graph, providing a more comprehensive understanding of classes. The authors propose a spectral approach for zero-shot node classification (SpeAr). Establishing an approximate relationship between minimizing the spectral contrastive loss and performing spectral decomposition on the graph, thereby enabling effective node characterization through loss minimization. The class prototypes are iteratively refined based on the learned node representations, initialized with the semantic vectors. Experiments verify the effectiveness of the SpeAr, which can further alleviate the bias problem.

**Strengths:**

- Overall, I thoroughly enjoyed reading this paper and have a highly favorable impression.

- Learning clustering information for unlabeled nodes represents a pivotal research direction.

- The novel utilization of spectral clustering and the development of a spectral contrast loss function constitute a noteworthy contribution.

- The representation is concise and coherent. The proposed approach demonstrates remarkable efficacy.

**Weaknesses:**

- High computational complexity. Insufficient experimentation, what it will be if the model is not pre-trained.

- This paper provides a theoretical analysis and there is potential for further elaboration and clarification of the intrinsic correlation between theory and practical performance.

**Questions:**

1. What are the fundamental differences between traditional zero-shot learning (ZSL) and zero-shot node learning? Furthermore, is there a potential for applying the SpeAr method to zero-shot learning for image classification tasks?

2. In the section 3.5, pre-training is highlighted as a crucial step. Could you elaborate on the necessity of this pre-training phase for the proposed method, and what implications it may have if it were omitted?

3. Could you provide a more detailed explanation of how the proposed approach leads to improved performance, grounded in the theoretical framework outlined in Theory 3.1?

4. The paper mentions refining the prototype update process by selecting the top-s nodes with the highest scores. Could you elaborate on the methodology for determining the optimal value of the hyper-parameter s?

**Limitations:**

From my opinion, there is no potential negative societal impact of their work.

---

> ### Author Rebuttal · Authors · 2024-08-06
>
> Thank the reviewer for carefully reading and for the valuable comments. We greatly appreciate the time and effort you have taken to provide such thoughtful feedback. Below is a point-by-point response to the comments.
>
> **> Response to W1: High computational complexity**
>
> Given the shared concerns expressed by Reviewer rWPM and yourself, we have formulated a comprehensive and unified response to this comment within our **General Response**. We respectfully request that you consult this General Response for our full explanation and clarification of this issue.
>
> **> Response to W1 and Q2: The important of pre-trained**
>
> The first phase is our pre-training phase. In the Ablation Study (Section 4.3) of the main body, we validate the model without the first phase (Model2). As shown in the following table, the Model2's results decline across three datasets.
>
> |Dataset|	Model2|	SpeAr|
> | - | -|-|
> |Cora|	52.42|	60.48|
> |Citeseer	|48.15|	59.72|
> |C-M10M|	50.27|	54.22|
>
> In SpeAr, we use the pre-trained model mainly for several reasons:
>
> - Improve the training efficiency of the second phase: In SpeAr, the first phase leverages the unsupervised contrastive loss to train the model. In the second phase, the focus is on training only the final layer of the network. This approach improves training efficiency by reducing GPU memory usage and significantly decreasing the number of gradient updates in the second stage.
>
> - Simulate real-world workflows: In the industry, it is standard practice to first train a model using self-supervised loss on unlabeled data. In the first phase, the model can take advantage of the unlabeled data to learn useful representations from the data and enhance the model's understanding of the data.
>
> **> Response to W2 and Q3: The relationship between theory and practical performance**
>
> Given that you share the same concerns as Reviewer 7tvx, we have addressed the relationship between theory and performance comprehensively in **General Response**. We kindly request you to refer to the General Response for our detailed response, which we hope will clarify your doubts. Your patience is greatly appreciated.
>
> **> Response to Q1: Differences between traditional zero-shot learning (ZSL) and zero-shot node learning**
>
> While both traditional ZSL and zero-shot node classification (ZNC) aim at the recognition of unseen class samples, there are some differences:
>
> 1. Data type:
>
> - ZSL: Images, text, or audio data is independent identical distribution.
>
> - ZNC: Graph data, nodes form a network structure through connectivity (edges).
>
> 2. Feature representation approaches:
>
> - ZSL: Features are usually extracted by pre-trained feature extractors (e.g. ResNet).
>
> - ZNC: Feature representation not only considers the nodes attributes but also integrates the neighbor information of the nodes and the global graph structure.
>
> 3. Semantic migration pathway:
>
> - ZSL: Use semantic information (e.g., word vectors, attribute vectors) to transfer semantic knowledge from seen classes to unseen ones.
>
> - ZNC: Rely on not only semantic information but also relationships between nodes for semantic migration.
>
> To summarize, traditional ZSL is significantly different from ZNC in terms of data types, feature representations, and semantic migration pathways.
>
> In future work, applying the SpeAr to ZSL for image classification tasks is feasible. The critical aspect is constructing the neighbor relationships between samples, which can be achieved through feature similarity or image augmentation strategies.
>
> **> Response to Q4: The choice of the hyperparameter $s$**
>
> For the choice of the hyperparameter $s$, it is set to 1000 across all three datasets. In SpeAr, we found that if $s$ is set too small, its effect on prototype updates is not significant. As shown in the table below, when $s$=100, the results across the three datasets are poor. Based on the experimental results from all three datasets, we found that $s$=1000 achieved better performance consistently.
>
> |$s$	|Cora	|Citeseer	|C-M10M|
> |-|-|-|-|
> |100	|49.90	|54.32	|42.98|
> |200	|51.10	|56.93	|43.66|
> |300	|55.86	|57.98	|45.69|
> |400	|57.13	|58.40	|47.70|
> |500	|56.67	|58.87	|47.93|
> |600	|55.12	|59.86	|49.80|
> |700	|57.65	|58.00	|51.65|
> |800	|57.15	|**60.83**	|53.49|
> |900	|*58.51*	|59.47	|*54.01*|
> |1000	|**60.48**	|*59.72*	|**54.22**|

---

> > ### Author Response · Authors · 2024-08-11
> >
> > Thank you very much for your thorough review and insightful comments. We hope that our additional evaluations and rebuttal have addressed your primary concerns with our paper. We would really appreciate feedback as to whether there are any (existing or new) points we have not covered, and we would be happy to address/discuss them! Your comments are invaluable in helping us to refine and strengthen our work.

---

> > > ### Comment · Reviewer_HC2X · 2024-08-12
> > > **Responses**
> > >
> > > Thank you for your rebuttal. I'm satisfied with how you have addressed the time complexity concerns and clarified the relationship between theory and methodology in your paper. My original scores remain unchanged.
> > >
> > > Regarding time complexity, your validation makes sense. While it's true that the proposed method at epoch=1000 takes more time than other methods, the fact that it still outperforms them when the runtime is comparable is promising and shows potential.
> > >
> > > I think the proposed algorithm is applicable to both the zero-shot problem and other weakly-supervised challenges. I want to know where your research is going.
> > >
> > > Additionally, the adjacency matrix is a naturally existing and critical information of graph data. I'm curious if it could be leveraged to construct the spectral contrastive loss during pre-training. I'd like to see more details about this in your final version.

---

> > > > ### Author Response · Authors · 2024-08-12
> > > >
> > > > Thank you for your approval and response.
> > > >
> > > > **>Response to future work**
> > > >
> > > > In future work, we will focus on the following two points:
> > > >
> > > > 1. **Leveraging inter-class relationships to mine the clustering information inherent in unlabelled nodes.** In zero-shot node classification, the challenge of extracting clustering information is amplified when the unseen class closely mirrors the seen classes. However, a stark divergence between unseen and seen classes often leads to these unseen classes demonstrating a more potent capacity for self-clustering. Consequently, our focus is directed toward exploring and exploiting these similarities and differences between classes, aiming to prompt the model's efficacy in mining clustering information within unlabeled nodes.
> > > >
> > > > 2. **Broadening our model to weakly-supervised tasks**. Indeed, the versatility of our method allows for its application across a range of weakly-supervised tasks, including but not limited to few-shot node classification, semi-supervised node classification, and various other scenarios. Taking into account structural information to mine the inherent clustering information present in graphs represents a significant approach to tackling weakly supervised tasks.
> > > >
> > > > **>Response to whether or not to exploit the adjacency matrix in the pre-training phase.**
> > > >
> > > > That's an insightful question. It was indeed a factor we considered in the model's design. Nonetheless, the adjacency matrix was not incorporated during the pre-training phase for the reasons that follow:
> > > > 1. **Overfitting risk**: During the pre-training stage, the utilization of the adjacency matrix poses the risk of inducing the model excessively reliant on neighboring information, which could precipitate a descent into a local optimum. Should the model persist in its reliance on the adjacency matrix in the subsequent stage, it may find itself unable to break free from this local optimum, hindering its capacity to seek out a global solution, which could culminate in overfitting.
> > > >
> > > > 2. **Extending training time**: The infusion of excessive redundant information during the pre-training stage increases the risk of wasting computational resources and could substantially elongate the training period, consequently detracting from the overall efficiency. To avoid these pitfalls, we carefully selected the information used for the pre-training phase, excluding the adjacency matrix. We will append this aspect of our discussion in the final version.

---

### Official Review · Reviewer_rWPM · 2024-07-16

**Soundness:** 2
**Presentation:** 3
**Contribution:** 3
**Rating:** 5
**Confidence:** 3

**Summary:**

This paper proposes a spectral approach for zero-shot node classification (ZNC) that addresses prediction bias based on node representation learning technique. It optimizes node representations by a two-stage training method with spectral contrastive loss and class prototypes, in which the class prototypes are initialized using document embeddings related to the class. Experiments have been conducted to verify the effectiveness of the proposed method on ZNC tasks.

**Strengths:**

1. It is intuitive and reasonable to address the prediction bias in zero-shot node classification using the cluster information in the graph.
2. Experiments have been conducted on three benchmarks and the results show the effectiveness of the proposed method compared with other ZNC method.
3. The paper is well organized and explains the proposed method very clearly.

**Weaknesses:**

1. The design of the spectral contrastive loss function relies on previous work [29] with only slight variations in the specific form of positive and negative pairs.
2. The proposed method might be inefficient on large-scale datasets since the spectral contrastive loss requires multiple loss calculations.
3. It can be seen from Figure 2(b) and 2(c), the accuracy of the proposed method is sensitive to the hyperparameters $\alpha$ and $\beta$, which may require significant effort to make the method work on new datasets.
4. The experimental settings are unclear. For the sake of reproducibility, the values of hyperparameters of the model (e.g., the embedding dimension of the graph neural network) should be specified.

**Questions:**

1. For large-scale datasets, subgraph-based training leads to slower convergence. The authors should discuss the training times compared to other methods across all datasets in details.
2. Since the seen classes do not need to be classified during the test phase in the ZNC tasks, one of my concerns is that introducing too much seen class information during the train phase (e.g., reconfiguring the adjacency matrix and calculating spectral contrastive loss) may lead to overfitting and affect generalization to the unseen classes. Could the authors provide a discussion or explanation for how to address this issue?

**Limitations:**

The authors describe some limitations of the proposed method in Appendix.

---

> ### Author Rebuttal · Authors · 2024-08-06
>
> We are immensely grateful to you for recognizing the reasonability and presentation of our work. Your valuable suggestions inspire us to improve our work further.
>
> If you think the following response addresses your concerns, we would appreciate it if you could kindly consider raising the score.
>
> **> Replay to W1: Differences from the spectral contrastive loss in [29]**
>
> Our loss is designed based on the data and tasks, which is quite different from the loss in literature [29], as shown in the Table.
>
> | | Literature [29] | Ours |
> | - | -|-|
> | Data Format | Image Data | Graph Data |
> | Information Utilization | Unsupervised tasks, image augmentation | Label Information, nodes adjacency relationships |
> | Positive Pair Construction | Solely relying on augmented images as positive samples | (1) Relying on label information for labeled samples; (2) Relying on adjacency relationships for unlabeled samples|
> | Negative Pair Construction |  It may treat the same class of nodes as negative pairs | (1) Nodes with different labels form negative pairs; (2) Labeled and unlabeled nodes (No edges) are mutually treated as negative pairs |
> | Objective | Enhance the representation ability of images | (1) Improve the separability of seen classes; (2) Effectively mine the hidden clustering structures within the graph |
>
> Utilizing the spectral contrastive loss from [29] in ZNC is hindered by data and objectives. [29] targets image-level representations, establishing sample relationships via augmentation. Lacking labels and natural node adjacencies, [29] cannot effectively uncover positive correlations among similar nodes or negative ones between dissimilar nodes.
>
> It is important to note that our innovations are:
> 1. Combine the spectral contrastive learning with the learnable class prototypes to discover the implicit clustering information and realize the semantic migration, thus further alleviating the bias problem.
> 2. Make full use of labeling and adjacency to design node spectral contrastive loss mining the implicit clustering information in unlabeled nodes (**Reviewer HC2X also notes this**).
> 3. Theoretically, we derive an approximate relationship between the obtained node embeddings and the feature vectors obtained from spectral decomposition.
>
> **> Replay to W2 & Q1: The efficiency and training time**
>
> Given that your concerns align with Reviewer HC2X, we have responded to this issue in the **General Response**. We sincerely appreciate your time and patience, and kindly encourage you to navigate to that section for our comprehensive response.
>
> **> Replay to W3: The sensitivity of the hyperparameters**
>
> We have set guidelines for parameter selection that efficiently guide in choosing suitable hyperparameters and facilitate swift model tuning for new data.
>
> **Hyperparameters $\alpha$ and $\beta$**
>
> We analyze the $\alpha$ and $\beta$ across three datasets (**Figure 1 in the additional rebuttal PDF**). The $\alpha$ and $\beta$ regulate the contribution of labeled and unlabeled samples in overall loss. The results reveal a notable trend: when the value of $\beta$ is larger than $\alpha$, the model generally exhibits better performance, with this advantage being particularly prominent when $\beta = 1$.
>
> The conclusions meet our expectations.  A larger $\beta$ can reduce overfitting to seen classes and better exploit the clustering information embedded in unlabeled nodes. Thus, the parameters need to satisfy $\beta > \alpha$. This suggests that the $\alpha$ and $\beta$ are not sensitive to some extent.
>
> **Hyperparameters $q$**
>
> When a sample's pseudo-label probability exceeds the threshold $q$, it updates the class prototypes. The choice of $q$ depends on model and dataset: low $q$ risks noisy pseudo-labels, degrading performance; high $q$ ensures accuracy but limits unlabeled data utilization [1-3].
>
> Fortunately, SpeAr achieves satisfactory performance across datasets with $q$=0.5 (Figure 2(c) in Section 4.4). Even on Cora, our result (55.49) at $q$=0.5 significantly surpasses SOTA (48.43), validating the effectiveness of SpeAr. The strategies for handling new datasets are:
>
> 1. Set $q$=0.5, assigning lower weights to samples during prototype updates to mitigate noise.
>
> 2. Select $q$ using the validation set (**Figure 2 in the additional rebuttal PDF**). Across three datasets, as $q$ varies, the accuracy changes in both the test and validation sets are roughly consistent, confirming the strategy's effectiveness and reliability. This will be discussed in the modified version.
>
> 3. In the future, we will focus on advanced threshold selection, including adapting thresholds to class difficulty and dynamically updating them based on model learning.
>
> [1] Fixmatch: Simplifying semi-supervised learning with consistency and confidence
>
> [2] Boosting semi-supervised learning by exploiting all unlabeled data
>
> [3] SemiReward: A General Reward Model for Semi-supervised Learning
>
> **> Replay to W4: Inadequate experimental settings**
>
> We give all the hyperparameters involved in our SpeAr (**Table 1 in the additional rebuttal PDF**).
>
> **> Replay to Q2: Caution in using seen classes to prevent overfitting**
>
> We accounted for this in the initial manuscript (Section 4.4) by adjusting the weights of $\alpha$ and $\beta$ to further alleviate overfitting issues.
>
> First, the seen class information serves as a pivotal semantic bridge during the training phase, facilitating SpeAr's ability to capture and transmit the deep semantic ties between classes, enhancing the model's capacity for generalization towards unseen classes.
>
> Second, we adjusted the weights of seen and unseen classes within the spectral contrastive loss to mitigate the overfitting to seen classes, setting $\beta$ to be greater than $\alpha$. The adjustment also improves the model's adaptability to unseen classes.
>
> Recognizing that our manuscript's lack of detailed explanation may have led to confusion, we will add a detailed discussion in the revised version.

---

> > ### Author Response · Authors · 2024-08-11
> >
> > Thanks very much for your thorough review and insightful comments. We hope that our additional evaluations and rebuttal have addressed your primary concerns with our paper. We would really appreciate feedback as to whether there are any (existing or new) points we have not covered, and we would be happy to address/discuss them! Your comments are invaluable in helping us to refine and strengthen our work.

---

> > > ### Comment · Reviewer_rWPM · 2024-08-14
> > >
> > > Thanks for your detailed feedback. I think some of my concerns on the difference from [29], efficiency and hyperparameters have been addressed. Thus, I will raise my score.

---

> > > > ### Author Response · Authors · 2024-08-14
> > > >
> > > > We extend our heartfelt appreciation for your inspiring decision. We will meticulously incorporate the relevant discussion from the responses into the revised version.

---

### Author Rebuttal · Authors · 2024-08-06

## **General Response**

We sincerely appreciate the reviewers for their valuable and constructive comments.

We are honored to see that the reviewers recognized the novelty (HC2X, 7tvx), reasonability (rWPM, 7tvx), and significant contributions (HC2X, 7tvx) of our framework. Several reviewers appreciated the clear motivation (rWPM, HC2X, 7tvx) and structure (7tvx) of this paper. Reviewer HC2X acknowledged that our method is simple and effective, outperforming existing methods (rWPM, HC2X, 7tvx) in zero-shot node classification tasks. Reviewer 7tvx commended the theoretical depth of our paper. Additionally, reviewers (rWPM, HC2X) praised our writing.

We have addressed the comments and concerns of each reviewer in our individual responses.

Here are our answers to the common questions:

- Answers to the questions of Reviewer rWPM and Reviewer HC2X about efficiency and computational complexity.

- Answers to the questions of Reviewer HC2X and Reviewer 7tvx about the relationship between theory and performance.

**> Response to Reviewer rWPM and Reviewer HC2X: The efficiency and computational complexity**

Our SpeAr involves computing spectral contrastive loss across all nodes and updating prototypes, which increases computational requirements.

For the large dataset ogbn-arxiv, we adopt the “minibatch + sampling” strategy. Specifically,  the minibatch size is 4096,  and each node samples only a small and fixed number of neighboring nodes, which effectively avoids the risk of memory overflow and also ensures the time efficiency of SpeAr.

In addition, by reducing the number of epochs (1000 $\rightarrow$ 100), the running time of SpeAr is comparable to that of DBiGCN (epoch=10000). However, it is worth noting that the accuracy of SpeAr still significantly outperforms that of DBiGCN. That's because SpeAr mines and utilizes more information. Next, we first give the running time of SpeAr under 1000 epochs for zero-shot node classification (ZNC) in Table 1. Then, we show its running time under 100 epochs in Table 2. Finally, we present the accuracy of SpeAr under 100 epochs in Table 3.

Table 1: Running time of the DGPN, DBiGCN, our SepAr (epoch=1000), First phase, and Second phase (epoch=1000) for ZNC (s means seconds, h means hours).
| | DGPN | DBiGCN | SpeAr (epoch=1000) | First phase | Second phase (epoch=1000) |
| - | -|-| -|-|-|
| Cora | 37s | 138s | 739s | 88s | 651s |
| Citeseer | 90s | 237s | 795s | 103s | 692s |
| C-M10M | 51s | 228s | 854s | 146s |708s |
| ogbn-arxiv |6.2h |26.7h | 86.1h|9.4h |76.7h |

Table 1 compares the runtime of DGPN, DBIGCN, and SpeAr, leveraging the Python implementations from their respective authors. On a unified platform, SpeAr's execution yields higher time consumption. In each epoch, SpeAr requires matrix calculations, loss calculations, and prototype updates.

Remarkably, Table 2 shows that scaling down to 100 epochs significantly reduces runtime for SpeAr, while Table 3 demonstrates that SpeAr maintains accuracy superior to existing methods. This underscores SpeAr's ability to attain commendable performance with fewer epochs due to its comprehensive utilization of information.

Table 2: Running time of the DGPN, DBiGCN, our SepAr (epoch=100), First phase, and Second phase (epoch=100).
| | DGPN | DBiGCN | SpeAr (epoch=100) | First phase | Second phase (epoch=100) |
| - | -|-| -|-|-|
| Cora | 37s | 138s | 169s | 88s | 81s |
| Citeseer | 90s | 237s | 189s | 103s | 86s |
| C-M10M | 51s | 228s | 210s | 146s | 64s |
| ogbn-arxiv | 6.2h|26.7h |20.2h |9.4h | 10.8h|

Table 3: Zero-shot node classification accuracy (%).
| | DGPN | DBiGCN | SpeAr (epoch=100) |
| - | -|-| -|
| Cora | 33.78 | 45.14 | **55.46** |
| Citeseer | 38.02 | 40.97 | **52.78** |
| C-M10M | 41.98 | 45.45 | **50.02** |
| ogbn-arxiv | 22.37 | 21.40 | **26.23**|

In summary, by adopting the "minibatch + sampling" strategy, we can improve the efficiency of SpeAr on large-scale datasets. We are pleasantly surprised to find that even if the training time is shortened, the SpeAr performance is still competitive. We maintain that, despite an increase in the algorithm's runtime (when epoch=1000), the importance of mining more valuable information significantly outweighs this aspect. We eagerly await your thoughts on this idea and look forward to further discussions on this matter.

**> Response to Reviewer HC2X and Reviewer 7tvx: The relationship between theory and practical performance**

1. Spectral decomposition can mine the intrinsic clustering structure of unlabeled nodes: ZNC aims to identify nodes of classes unseen during the training process. Leveraging the inherent cluster information of unlabeled nodes is essential for enhancing the model's ability to recognize and understand unseen classes. Therefore, we combine spectral analysis with learnable category prototypes to reveal the intrinsic clustering structure in the graph.

2. In SpeAr, the reshaped adjacency matrix $A$ includes label information and node adjacency relationships. SpeAr is actually a method of factorizing the adjacency matrix.

3. Based on 1 and 2, Theory 3.1 derives the spectral contrastive loss on the graph from the spectral decomposition. The spectral contrastive loss contains several losses, each acting as follows:

   $\mathcal{L}_{1}$ ensure intra-class compactness of labeled nodes.

   $\mathcal{L}_{2}$ mines clustering information of unlabeled nodes using adjacency.

   $\mathcal{L}_{3}$ guarantees interclass separability of labeled nodes.

   $\mathcal{L}_{4}$ constraints the separability of seen and unseen classes.

   $\mathcal{L}_{5}$ treats all remaining unlabeled node pairs (except for node pairs identified as positive nodes in the loss
   $\mathcal{L}\_{2}$ ) as negative pairs.

Spectral contrastive loss can exploit the structure of the graph data to mine cluster information in unlabeled nodes and ensure the distinguishable between classes in the feature space.

---

### Decision · Program_Chairs · 2024-09-25

**Decision:**

Accept (poster)

**Comment:**

The reviewers collectively agree that the proposed spectral approach for zero-shot node classification (ZNC) is a significant and novel contribution that effectively addresses prediction bias by leveraging spectral clustering and class prototypes. Despite some concerns about computational complexity and sensitivity to hyperparameters, the method's theoretical foundation, clear presentation, and demonstrated effectiveness in experiments are highly appreciated. Based on these strengths, I recommend accepting the paper.